# Estimating mutation rates under heterogeneous stress responses

**Lucy Lansch-Justen** [1]*, **Meriem El Karoui** [2,3,4], **Helen K. Alexander** [1,3]*

**1** Institute of Ecology and Evolution, School of Biological Sciences, University of Edinburgh, Edinburgh, Scotland, United Kingdom, **2** Institute of Cell Biology, School of Biological Sciences, University of Edinburgh, Edinburgh, Scotland, United Kingdom, **3** Centre for Engineering Biology, University of Edinburgh, Edinburgh, Scotland, United Kingdom, **4** Bacterial Systems Biology and Anti Microbial Resistance, Laboratoire de Biologie et Pharmacologie Appliquée, École Normale Supérieure Paris-Saclay, Gif-sur-Yvette, France

* lucy.lanju@googlemail.com (LLJ); helen.alexander@ed.ac.uk (HKA)

## Abstract

Exposure to environmental stressors, including certain antibiotics, induces stress responses in bacteria. Some of these responses increase mutagenesis and thus potentially accelerate resistance evolution. Many studies report increased mutation rates under stress, often using the standard experimental approach of fluctuation assays. However, single-cell studies have revealed that many stress responses are heterogeneously expressed in bacterial populations, which existing estimation methods have not yet addressed. We develop a population dynamic model that considers heterogeneous stress responses (subpopulations of cells with the response *off* or *on*) that impact both mutation rate and cell division rate, inspired by the DNA-damage response in *Escherichia coli* (SOS response). We derive the mutant count distribution arising in fluctuation assays under this model and then implement maximum likelihood estimation of the mutation-rate increase specifically associated with the expression of the stress response. Using simulated mutant count data, we show that our inference method allows for accurate and precise estimation of the mutation-rate increase, provided that this increase is sufficiently large and the induction of the response also reduces the division rate. Moreover, we find that in many cases, either heterogeneity in stress responses or mutant fitness costs could explain similar patterns in fluctuation assay data, suggesting that separate experiments would be required to identify the true underlying process. In cases where stress responses and mutation rates are heterogeneous, current methods still correctly infer the effective increase in population mean mutation rate, but we provide a novel method to infer distinct stress-induced mutation rates, which could be important for parameterising evolutionary models.

## Author summary

How does environmental stress, especially from antibiotics, affect mutation rates in bacteria? This question has often been examined by estimating mutation rates using fluctuation assays, an experiment dating back to Luria and Delbrück in the 1940s. In this study, we consider variation in stress responses within bacterial populations, as revealed by recent

**Data Availability Statement:** The complete annotated documentation of the computational analyses of this study is archived on Zenodo, https://doi.org/10.5281/zenodo.11174801.

**Funding:** This work was supported by the UKRI Biotechnology and Biological Sciences Research Council (BBSRC) grant number BB/T00875X/1 and a University of Edinburgh Principal's Career Development PhD Scholarship to LLJ, a Wellcome Trust Investigator Award 205008/Z/16/Z to MEK, and a Royal Society University Research Fellowship URF/R1/191269 to HKA. The funders had no role in study design, data collection and analysis, decision to publish, or preparation of the manuscript.

**Competing interests:** The authors have declared that no competing interests exist.

single-cell studies, which is neglected in currently available mutation-rate estimation methods. Our approach involves a population dynamic model inspired by the DNA-damage response in *E. coli* (SOS response). It accounts for a subpopulation with high expression of the stress response, which increases the mutation rate and decreases the division rate of a cell. We use computer simulations to generate synthetic fluctuation assay data. Notably, we find that over a wide range of scenarios, existing models and our heterogeneous-response model cannot be distinguished using fluctuation assay data alone. This emphasises the need for separate experiments to uncover the true underlying processes. Nevertheless, when stress responses are known to be heterogeneous, our study offers a novel method for accurately estimating mutation rates specifically associated with the high expression of the stress response. Uncovering the heterogeneity in stress-induced mutation rates could be important for predicting the evolution of antibiotic resistance.

## Introduction

Bacteria are commonly exposed to adverse conditions, such as starvation, sub-optimal temperatures or toxins, including antibiotics. To cope with these conditions, bacteria have evolved a range of stress responses that enhance viability under stress, often at the expense of a lower growth rate. Some of these response pathways also increase mutagenic mechanisms by, for example, increasing the expression of error-prone DNA polymerases or down-regulating error-correcting enzymes [1, 2]. It has been proposed that this so-called 'stress-induced mutagenesis' (SIM) in bacterial cells could accelerate the evolution of populations that are poorly adapted to their environment [3–6]. Consequently, inhibiting bacterial stress responses has been suggested to prevent antibiotic resistance evolution and gained some experimental support [7–9].

Several studies report increased mutation rates in bacterial populations exposed to sublethal antibiotic concentrations [8, 10–15]. These mutation rates have been typically measured with fluctuation assays. This experiment (see, for example, [16] for a protocol) involves inoculating several parallel cultures at a small population size and growing them under permissive conditions for several hours, typically overnight. During this *growth phase*, mutations occur randomly, and the experiment is designed to minimise selection on mutant cells. Subsequently, each culture is plated on strong selective media such that only mutant cells can grow and form a colony. The mutation rate to the chosen selective marker is estimated from the distribution of the number of mutant colonies on the plates, the *mutant count distribution*; see [17] for a summary of estimation methods. The experiment is repeated to quantify the mutation-rate increase associated with stress, by exposing the cultures to a stressor during the growth phase. Then, the *stress-induced* mutation rate is estimated and compared with the mutation rate under permissive conditions. However, stress impacts the growth of bacterial cells in several ways, which are neglected in commonly applied estimation methods, potentially leading to biased estimates of the mutation rate. For instance, increased cell death leads to overestimating the mutation rate [18]. Another effect that has not yet been addressed is within-population heterogeneity in stress responses.

In recent years, single-cell experiments have revealed extensive heterogeneity in the expression of stress responses in bacterial populations [8, 19–28]. Heterogeneity can arise for various reasons, including stochastic expression of genes involved in stress responses, especially where the corresponding proteins are initially present in small numbers [20–22], phenotypic variability in the stability of key regulators [25], or micro-environmental variation in cell-to-cell

interactions [28]. Positive and negative feedback loops are common features of stress response regulatory networks, which can generate, amongst other features, cell-to-cell variation [29]. In some cases, a subpopulation of cells showing elevated stress responses has been directly associated with a higher rate of DNA mismatches or higher mutant frequency [8, 20–22, 24, 26].

In addition to mutagenic mechanisms, stress responses can alter cell division and death rates. For example, the widely studied SOS response, which leads to the transcriptional induction of approximately 40 genes after exposure to DNA damage, involves inhibition of cell division, filamentation and induction of error-prone DNA polymerases that could increase mutation rate [30, 31]. Single-cell studies using fluorescent reporters for the SOS response in *E. coli* have revealed that its expression is highly heterogeneous. Under certain conditions, a subpopulation of cells with a very high level of SOS compared to the rest of cells with lower expression levels has been observed, and this heterogeneity can be approximated as a bimodal response [19, 21, 27]. Overall, heterogeneously expressed stress responses are, therefore, likely to impact both bacterial population dynamics and mutational input during the growth phase of a fluctuation assay, and it is unclear whether estimation methods that neglect heterogeneity in stress responses produce reliable results.

In this study, we present a population dynamics model that considers within-population heterogeneity in stress responses. Motivated by the SOS response, we describe two discrete subpopulations of cells, where high expression of the stress response is associated with both a higher mutation rate and a lower division rate than in cells with low expression. We derive the resulting mutant count distribution in the total population and implement maximum likelihood estimation of the mutation-rate increase associated with the induction of the stress response. We test the performance of our method using stochastic simulations of fluctuation assays under permissive and stressful conditions, including robustness to biologically realistic model deviations such as mutant fitness costs and cell death. We also apply formal model comparison to assess whether within-population heterogeneity could be detected from fluctuation assays alone.

## Model and methods

Studying stress-induced mutagenesis with fluctuation assays requires a pair of experiments: one with a growth phase under permissive conditions (as a baseline for comparison) and one under 'stressful' conditions, where a stressor such as a low dose of antibiotic (which is supposed to induce a mutagenic stress response in the cells) is added during the growth phase. In addition to performing the experiments, researchers have to decide on a mathematical model of the underlying dynamics, including the population dynamics of non-mutants and mutants during the growth phase, and how these dynamics change under exposure to stress. This then allows them to estimate the model parameters, most importantly mutation rates, and assess increases in mutation rates due to stress. Many studies of SIM to date, for example [8, 11, 14], have implicitly assumed that the stress response is homogeneous, i.e. the stressful condition results in a population-wide elevation of the mutation rate. In contrast, our new model considers within-population heterogeneity in stress responses and mutation rates.

In the following, we recap what we call the *standard model* used in a classical fluctuation assay and extensions particularly relevant to stress. Then, we formalise the *homogeneous-response model*, a version of which is considered in the aforementioned studies of SIM, and introduce our new *heterogenenous-response model* with a detailed description of the population dynamic model under heterogeneous stress responses and the derivation of the resulting mutant count distribution. Next, we describe our model fitting and parameter estimation approach using maximum likelihood estimation and summarise the inference parameters.

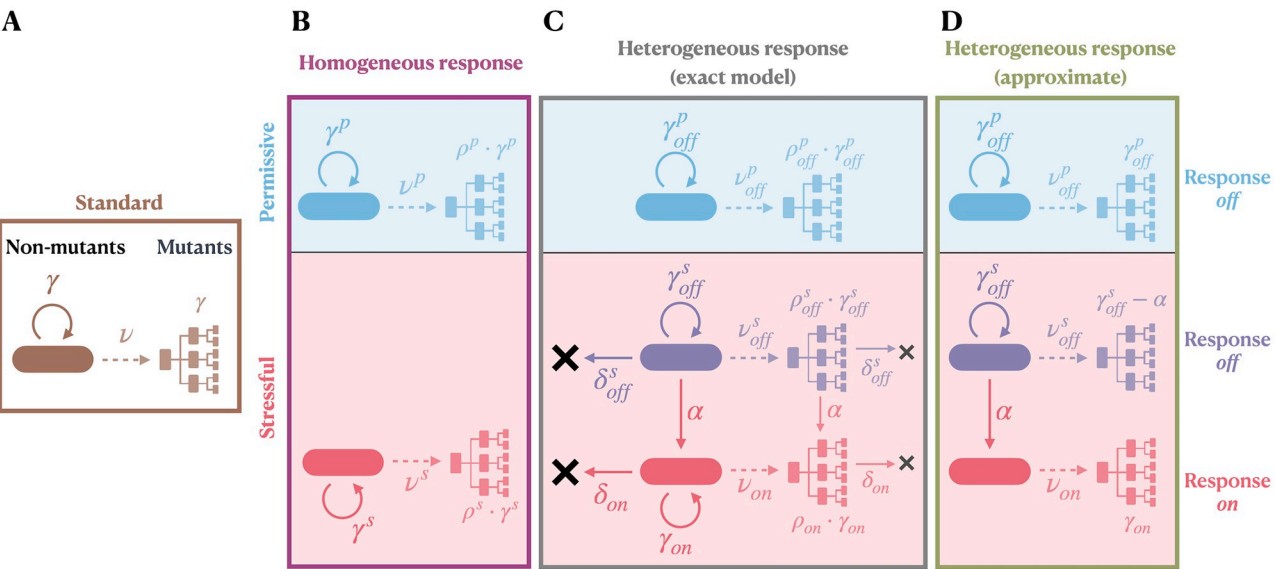

**Fig 1. Schematic illustrating the standard model of fluctuation assays and models of homogeneous and heterogeneous responses to stress.** In the standard model (**A**), non-mutants are assumed to grow exponentially (rate $\gamma$), mutations to arise randomly (rate $\nu$ per cell per unit time), and mutants to divide stochastically (birth rate $\gamma$). In models of homogeneous stress responses (**B**), it is assumed that both fluctuation assays under permissive (superscript $p$, light blue) and stressful (superscript $s$, light red) conditions can be described by the standard model, with optional differential fitness of mutants compared to non-mutants (factor $\rho$). In our model of heterogeneous stress responses, on the other hand, we assume that the induction of the stress response (rate $\alpha$) results in the separation into two subpopulations: response-*off* (subscript *off*, dark purple) and response-*on* (subscript *on*, red). When simulating under the heterogeneous-response model, we use the exact model (**C**), optionally extended by cell death (rate $\delta$) and differential mutant fitness ($\rho$), where explicitly specified. For inference, we fit the approximate heterogeneous-response model (**D**). For the homogeneous-response model, we use the same version for both simulation and inference.

Finally, we describe the simulations to generate synthetic mutant count data, and how we evaluate the estimation methods using these data.

For schematics of the models used in simulation and inference, see Fig 1. The complete documentation of the computational methods can be found in the **README** at https://github.com/LucyL-J/Quantifying-SIM.

## The standard model of fluctuation assays and extensions relevant to stress

Classically, fluctuation assays have been described using what we refer to as the *standard model* (Fig 1A). In the standard model, the non-mutant population is assumed to grow exponentially during the growth phase, while the occurrence of mutations and the division of mutant cells are treated stochastically. On selective plates, it is assumed that every mutant cell (but no non-mutant cell) forms a visible colony. Many extensions of this standard model have been developed, briefly reviewed, for example, in [32]. Here, we describe two extensions particularly relevant to stress: accounting for cell death and allowing mutant cells to have a different fitness than non-mutant cells during the growth phase. The latter can become important when resistance allowing growth on the selective plates also confers an advantage to the stressor (for example, due to cross-resistance) or when mutants carry a fitness cost. Together, these two extensions result in the following population dynamic model. The non-mutant population grows exponentially,

$$N(t) = N_i \mathrm{e}^{\lambda t} \qquad (1)$$

with initial population size $N_i$ and population growth rate $\lambda$. Mutations occur according to a

time-inhomogeneous Poisson process with rate $vN(t)$. Note that $v$ describes the mutation rate to the phenotype of interest selected on the plates in the fluctuation assay (mutations per cell per unit time, also called instantaneous mutation rate). The dynamics of each mutant cell $M$ are captured by a continuous-time linear birth-death process [33] with birth rate $b$ and death rate $d$:

$$\begin{cases} M \rightarrow MM, & \text{rate } b \\ M \rightarrow \varnothing, & \text{rate } d, \end{cases} \qquad (2)$$

implying that mutants have a different fitness than non-mutants if $b - d \neq \lambda$.

For such dynamics, defining the per-generation mutation rate as $\mu := \frac{v}{\lambda}$, the differential fitness of mutants as $\rho := \frac{b-d}{\lambda}$ and the extinction probability of mutants as $\epsilon := \frac{d}{b}$, the resulting distribution of the number of mutants when the population reaches a final population size $N_f$ has been derived [34]: Assuming $N_f \gg N_i$ (neglecting initial population size effects), the probability-generating function (PGF) $G(z)$, a mathematically-convenient representation of the mutant count distribution, is given by

$$G(z) = \exp\left[ -\mu N_f (1 - \epsilon) F \left( \begin{matrix} 1, \dfrac{1}{\rho} \\ 1 + \dfrac{1}{\rho} \end{matrix} ; \dfrac{z - \epsilon}{z - 1} \right) \right] \qquad (3)$$

with $F$ being the hypergeometric function. Note that $z$ is a dummy variable in the PGF with no physical meaning, and $G(z)$ does not directly give the probability of observing a specific mutant count, but the probabilities can be calculated from $G(z)$ [34]. In the case where mutants have the same fitness as non-mutants and do not undergo cell death, the equation simplifies to:

$$G(z) = \exp\left[ \mu N_f \frac{(1 - z)\log(1 - z)}{z} \right], \quad \text{for } \rho = 1, \, \epsilon = 0 \qquad (4)$$

## Formalisation of the homogeneous-response model

The *homogeneous-response model* assumes that stress and stress responses impact mutation, division and death rates on a population-wide level. This implies that the dynamics under stressful conditions can, as under permissive conditions, be captured by the standard model (with optional extensions) as described above, simply substituting different parameter values (Fig 1B). Under permissive conditions (parameters denoted with a superscript $p$), assuming no cell death, the non-mutant population grows exponentially, $n^p(t) = n^p(0)e^{\gamma^p t}$, with division rate $\gamma^p$; mutations occur at rate $v^p n^p(t)$ and mutants develop according to a pure birth process with rate $\rho^p \cdot \gamma^p$.

Under stressful conditions (parameters denoted with a superscript $s$), the population grows as $n^s(t) = n^s(0)e^{(\gamma^s - \delta^s)t}$ with a different growth rate caused by a change in division rate $\gamma^s$ or a non-zero death rate $\delta^s$ or both. Mutations also occur at a different rate $v^s n^s(t)$, and the dynamics of mutants are given by a birth-death process with birth rate $\rho^s \cdot \gamma^s$ and death rate $\delta^s$.

Therefore, the stress response results in a population-wide change in the per-division mutation rate, $\mu^p = \frac{v^p}{\gamma^p} \rightarrow \mu^s = \frac{v^s}{\gamma^s}$; potentially the differential fitness of mutants, $\rho^p \rightarrow \rho^s$; and potentially a non-zero extinction probability of mutants, $\epsilon^s = \frac{\delta^s}{\gamma^s}$. The PGFs for the mutant count distributions under permissive and stressful conditions in the homogeneous-response model

are thus given by

$$
G^p_{\text{hom}}(z) \quad = \exp\left[-\mu^p N^p_f F\left(\begin{array}{c} 1, \dfrac{1}{\rho^p} \\ 1+\dfrac{1}{\rho^p} \end{array}; \dfrac{z}{z-1}\right)\right]
\tag{5}
$$

$$
G^s_{\text{hom}}(z) \quad = \exp\left[-\mu^s N^s_f(1-\epsilon^s) F\left(\begin{array}{c} 1, \dfrac{1}{\rho^s} \\ 1+\dfrac{1}{\rho^s} \end{array}; \dfrac{z-\epsilon^s}{z-1}\right)\right].
\tag{6}
$$

By applying standard mutation-rate estimation methods to both the fluctuation assay under permissive and the one under stressful conditions, studies of SIM to date have implicitly applied such a homogeneous-response model.

## Detailed description of the heterogeneous-response model

In contrast, our *heterogeneous-response model* considers within-population heterogeneity in the expression of the stress response under stressful conditions. Specifically, we suppose the population can be divided into two subpopulations: one with a low expression level of the stress response (here referred to as response switched *off*, even if strictly speaking the response is not fully *off* but very low) and the other with a high expression level (here referred to as response switched *on*). Each sub-population is associated with its own mutation rate and division rate. We adopt most of the same assumptions of the standard model while focusing on the specific effect of within-population heterogeneity upon induction of stress responses (Fig 1C).

Under permissive conditions, we assume that all cells have the response switched *off*, neglecting any stochastic switching in the absence of a stressor, and therefore, continue to use the standard model (with optional differential mutant fitness). In particular, the population grows exponentially, $n^p_{off}(t) = e^{\gamma^p_{off} t}$ with growth rate given by the division rate $\gamma^p_{off}$, mutations arise at rate $\nu^p_{off} n^p_{off}(t)$ and mutants develop according to a pure birth process with rate $\rho^p_{off} \cdot \gamma^p_{off}$. The PGF of the mutant count distribution is given by

$$
G^p_{\text{het}}(z) = \exp\left[-\mu_{off} N^p_f F\left(\begin{array}{c} 1, \dfrac{1}{\rho^p_{off}} \\ 1+\dfrac{1}{\rho^p_{off}} \end{array}; \dfrac{z}{z-1}\right)\right],
\tag{7}
$$

where $\mu_{off} := \dfrac{\nu^p_{off}}{\gamma^p_{off}}$ describes the *per-division* mutation rate, which equals the *per-generation* rate as, under permissive conditions, the population growth is solely determined by cell division.

**Population dynamic model under heterogeneous stress responses.** Upon exposure to stressful conditions, cells induce a stress response with a constant switching rate $\alpha$, leading to the emergence of a response-*on* subpopulation. Inducing the stress response alters the mutation rate of the cells but potentially also their division and death rates. We assume that, as long as the stress persists, cells do not switch the response *off* again.

**Non-mutant population dynamics.** We model the population sizes over time of the non-mutant response-*off* and response-*on* subpopulations, $n^s_{off}$ and $n_{on}$, respectively, with coupled

differential equations:

$$\dot{n}^s_{off} = (\gamma^s_{off} - \delta^s_{off} - \alpha)n^s_{off}, \tag{8}$$

$$\dot{n}_{on} = \alpha n^s_{off} + (\gamma_{on} - \delta_{on})n_{on}, \tag{9}$$

Here, $\gamma^s_{off}$ is the division rate of the response-*off* subpopulation under stress (which can be different than under permissive conditions, $\gamma^p_{off}$) and $\delta^s_{off}$ its death rate, $\gamma_{on}$ and $\delta_{on}$ are the division and death rates of the response-*on* subpopulation, and $\alpha$ is the switching rate. The solution to these equations is given by

$$n^s_{off}(t) = n^s_{off}(0)e^{(\gamma^s_{off} - \delta^s_{off} - \alpha)t} \tag{10}$$

$$n_{on}(t) = \frac{\alpha n_{off}(0)}{\gamma^s_{off} - \delta^s_{off} - \alpha - (\gamma_{on} - \delta_{on})}\left(e^{(\gamma^s_{off} - \delta^s_{off} - \alpha)t} - e^{(\gamma_{on} - \delta_{on})t}\right) + n_{on}(0)e^{(\gamma_{on} - \delta_{on})t} \tag{11}$$

with $n^s_{off}(0)$ and $n_{on}(0)$ denoting the initial numbers of response-*off* and response-*on* cells, respectively.

This approach assumes that the non-mutants, including the initially small response-*on* subpopulation, can be treated deterministically. We test the validity of this assumption using stochastic simulations (section A in S1 File): we simulate switching *on* of the response as a time-inhomogeneous Poisson process and the growth dynamics of the response-*on* subpopulation as a continuous-time linear birth-death process. Then, we compare the resulting population size with Eq 11. We find that deviations from the deterministic prediction are negligible for a wide range of switching rates and division rates of response-*on* cells and for zero and small initial sizes of the response-*on* subpopulation (Fig A in S1 File). Therefore, throughout the rest of this study, we treat non-mutants deterministically.

**Mutant population dynamics.**   We consider mutations in the response-*off* and the response-*on* subpopulation to occur according to time-inhomogeneous Poisson processes and treat the dynamics of the resulting mutants stochastically. Mutations arise in each subpopulation at rates $v^s_{off} n^s_{off}(t)$ and $v_{on}n_{on}(t)$, respectively. Importantly, mutation is not linked to cell division, but rather to chromosome replication. Expression of the SOS response, for example, inhibits cell division, but cells continue growing, leading to filamentation. Due to the continuation of chromosome replication, filamented cells may contain multiple chromosomes [35]. We neglect the possibility that these intracellular dynamics introduce heterogeneities amongst cells within the response-on subpopulation or over time and assume that the per-cell mutation rate ($v_{on}$) is constant. Experiments show that under prolonged low-level stress, multinucleated filamented cells can 'bud' viable, normal-sized progeny cells from their tips, some of which contain mutated chromosomes [35]. Although our model remains a simplification of this process, the experimental evidence indicates that response-*on* cells, even if largely non-dividing, can generate mutant offspring.

At the same time, since the selective agent on the plates is normally chosen to be unrelated to the stressor applied in the growth phase (e.g. two different antibiotics with no cross-resistance), we assume that mutation itself does not alter the stress response. For response-*off* cells, this implies that mutants can induce the response equivalently to non-mutants. Nonetheless, mutations might affect the fitness during the growth phase. Together, these assumptions result in a continuous-time two-type branching process describing the mutant response-*off* and response-*on* subpopulations, defined by the respective birth rates $\rho^s_{off}\gamma^s_{off}$ and $\rho_{on}\gamma_{on}$, respective

death rates $\delta_{off}^s$ and $\delta_{on}$, and switching at rate $\alpha$:

$$\begin{cases} M_{off} \rightarrow M_{off} \, M_{off}, & \text{rate } \rho_{off}^s \gamma_{off}^s \\ M_{off} \rightarrow \varnothing, & \text{rate } \delta_{off}^s \\ M_{off} \rightarrow M_{on}, & \text{rate } \alpha \\ M_{on} \rightarrow M_{on} \, M_{on}, & \text{rate } \rho_{on} \gamma_{on} \\ M_{on} \rightarrow \varnothing, & \text{rate } \delta_{on}. \end{cases} \tag{12}$$

On selective plates, where the stressor (which was applied during the growth phase) is no longer present, we assume that response-*on* mutant cells can resume division. Therefore, we continue to adopt the standard model assumption that every mutant cell forms a visible colony upon selective plating.

**Derivation of the mutant count distribution.**   To derive an analytical expression for the mutant count distribution, we make several approximations, resulting in the approximate heterogeneous-response model depicted in Fig 1D. First, we approximate Eq 11 as

$$\hat{n}_{on}(t) = \frac{\alpha n_{off}^s(0)}{\gamma_{off}^s - \delta_{off}^s - \alpha - (\gamma_{on} - \delta_{on})} e^{(\gamma_{off}^s - \delta_{off}^s - \alpha)t}. \tag{13}$$

This approximation is valid when the initial population size of the response-*on* subpopulation is comparably small, $n_{on}(0) \ll n_{off}(0)$, and its growth is slower than the growth of the response-*off* subpopulation, $\gamma_{on} - \delta_{on} \ll \gamma_{off}^s - \delta_{off}^s - \alpha$. As a consequence of this approximation, the total population grows exponentially with a *population growth rate* of

$$\lambda^s = \gamma_{off}^s - \delta_{off}^s - \alpha, \tag{14}$$

and the response-*on* subpopulation makes up a constant fraction of

$$f_{on}(t) = \frac{n_{on}(t)}{n_{on}(t) + n_{off}(t)} = \frac{\alpha}{\gamma_{off}^s - \delta_{off}^s - (\gamma_{on} - \delta_{on})} =: f_{on}^* \tag{15}$$

In the exact model given by the Eqs 10 and 11, the fraction of the response-*on* subpopulation changes with time until the stationary fraction $f_{on}^*$ is reached, but we assume that the fraction at the end of the growth phase, $f_{on}(t_f)$, is a good approximation of $f_{on}^*$ and for the rest of this study, we refer to it as simply the fraction of response-*on* cells $f_{on}$. Note that, even if response-*on* cells have zero division rate, the response-*on* subpopulation grows exponentially with the population growth rate $\lambda^s$ due to the induction of the stress response in response-*off* cells.

We define the *relative switching rate* as

$$\tilde{\alpha} := \frac{\alpha}{\gamma_{off}^s - \delta_{off}^s} \tag{16}$$

and the *relative fitness* of response-*on* compared to response-*off* cells under stressful conditions as

$$r_{on} := \frac{\gamma_{on} - \delta_{on}}{\gamma_{off}^s - \delta_{off}^s} \tag{17}$$

and thereby obtain

$$f_{on} = \frac{\tilde{\alpha}}{1 - r_{on}}. \tag{18}$$

This allows us to rewrite the population growth rate as a function of the division rate $\gamma^s_{off}$ and the extinction probability $\epsilon^s_{off} := \frac{\delta^s_{off}}{\gamma^s_{off}}$ of response-*off* cells, and the fraction $f_{on}$ and relative fitness $r_{on}$ of the response-*on* subpopulation

$$\lambda^s = \gamma^s_{off} - \delta^s_{off} - \alpha = \gamma^s_{off}(1 - \epsilon^s_{off})(1 - f_{on}(1 - r_{on})), \tag{19}$$

and to calculate the per-generation mutation rates of response-*off* and response-*on* cells

$$\frac{v^s_{off}}{\lambda^s} = \frac{v^s_{off}}{\gamma^s_{off}(1 - \epsilon^s_{off})(1 - f_{on}(1 - r_{on}))} = \frac{\mu_{off}}{(1 - \epsilon^s_{off})(1 - f_{on}(1 - r_{on}))} \tag{20}$$

$$\frac{v_{on}}{\lambda^s} = \frac{v_{on}}{\gamma^s_{off}(1 - \epsilon^s_{off})(1 - f_{on}(1 - r_{on}))} = \frac{\mu_{on}}{(1 - \epsilon^s_{off})(1 - f_{on}(1 - r_{on}))} \tag{21}$$

with $\mu_{on} := \frac{v_{on}}{\gamma^s_{off}}$. Importantly, we assume here that the per-division mutation rate of response-*off* cells is the same under stressful as under permissive conditions, $\mu_{off} = \frac{v^s_{off}}{\gamma^s_{off}} = \frac{v^p_{off}}{\gamma^p_{off}}$.

In an additional approximation to derive the mutant count distribution, we neglect the induction of the stress response in the mutants and assume that mutations have no fitness effect. For mathematical convenience, we consider switching as a reduction in the division rate of response-*off* mutants by $\alpha$ instead (birth rate equal to $\gamma^s_{off} - \alpha$). With this assumption, the dynamics of response-*off* and response-*on* mutant lineages are independent birth-death processes. For response-*on* cells, the relative fitness of mutants $\rho_{on}$ (relative to the population growth rate) can be expressed via

$$\rho_{on} = \frac{\gamma_{on} - \delta_{on}}{\lambda^s} = \frac{r_{on}}{1 - f_{on}(1 - r_{on})}. \tag{22}$$

With these approximations (see Fig 1D), the mutant counts in the response-*off* and response-*on* subpopulations are two independent stochastic processes, each following the standard model, with differential mutant fitness in the case of the response-*on* cells. Therefore, we can substitute the appropriate parameters into Eq 3 to obtain PGFs for the mutant counts, $G^s_{off}(z)$ and $G_{on}(z)$, respectively:

$$G^s_{off}(z) = \exp\left[\frac{\mu_{off}}{1 - f_{on}(1 - r_{on})}(1 - f_{on})N^s_f \frac{(1 - z)\log\left(\frac{1 - z}{1 - \epsilon^s_{off}}\right)}{z - \epsilon^s_{off}}\right] \tag{23}$$

$$G_{on}(z) = \exp\left[-\frac{\mu_{on}}{1 - f_{on}(1 - r_{on})}\frac{1 - \epsilon_{on}}{1 - \epsilon^s_{off}}f_{on}N^s_f F\left(1, \frac{1 - f_{on}(1 - r_{on})}{r_{on}}; 1 + \frac{1 - f_{on}(1 - r_{on})}{r_{on}}; \frac{z - \epsilon_{on}}{z - 1}\right)\right]. \tag{24}$$

Finally, the total mutant count distribution is given by the sum of the contributions of response-*off* and response-*on* subpopulations, with its PGF $G_{\text{het}}^s(z)$ given by the product $G_{off}^s(z) \cdot G_{on}(z)$.

In the case of $\gamma_{on} = 0$, the contribution to the mutant count from the response-*on* subpopulation follows a Poisson distribution, and (without cell death, implying $r_{on} = 0$) the PGF of total mutant count distribution reduces to

$$G_{\text{het}}^s(z) \quad = \exp\left[\overbrace{\mu_{off}N_f^s\frac{(1-z)\log(1-z)}{z}}^{\text{Response-}off\text{ contribution}} + \overbrace{\mu_{on}\frac{f_{on}}{1-f_{on}}N_f^s(z-1)}^{\text{Response-}on\text{ contribution}}\right] \tag{25}$$

$$= \exp\left[\mu_{off}N_f^s\left(\frac{(1-z)\log(1-z)}{z} + \frac{\mu_{on}f_{on}}{\mu_{off}(1-f_{on})}(z-1)\right)\right]. \tag{26}$$

$G_{\text{het}}^s(z)$ no longer depends on the mutation rate ($\mu_{on}$) and fraction ($f_{on}$) of response-*on* cells separately, but rather on the composite parameter

$$\mathcal{S} := \frac{\mu_{on}f_{on}}{\mu_{off}(1-f_{on})} \tag{27}$$

which gives the ratio of mutation supply coming from the response-*on* compared to the response-*off* subpopulation, with $\mathcal{S} = 0$ implying no heterogeneity in mutation rates.

For the purpose of comparison, we also define the increase in population mean mutation rate under stressful compared to permissive conditions:

$$\bar{M} := \frac{(1-f_{on})\mu_{off} + f_{on}\mu_{on}}{\mu_{off}}, \tag{28}$$

which is directly comparable to the increase in mutation rate in the homogeneous-response model, $\frac{\mu^s}{\mu^p}$, since $\mu^p$ and $\mu^s$ are population-wide rates.

Example mutant count distributions for the homogeneous and heterogeneous-response models are shown in Fig B in S1 File.

## Model fitting and parameter estimation using maximum likelihood

We use a maximum likelihood approach to estimate the model parameters from fluctuation assay data. For a given model (homogeneous or heterogeneous), we find the set of model parameters $\theta$ for which the observed mutant counts are most likely. Importantly, we consider mutant count data concurrently from a pair of fluctuation assays: one under permissive and the other under stressful conditions. In our heterogeneous-response model, there is at least one shared parameter between conditions ($\mu_{off}$); therefore, we consider the joint likelihood function. Note, however, that if there are no shared parameters between conditions (as is the case for some of the homogeneous-response models), the inference can be carried out separately.

We define a log-likelihood function

$$\ln \mathcal{L}(\theta \,|\, x^p, x^s) = \sum_{i=0}^{m^p} x^p(i) \ln[p_i^p(\theta)] + \sum_{i=0}^{m^s} x^s(i) \ln[p_i^s(\theta)] \tag{29}$$

as the natural logarithm of the probability of observing the mutant count distributions $x^p$ and

**Table 1. Parameters in the inference.**

| Symbol | Definition | Cond. | Model | In the inference |
|---|---|---|---|---|
| $N_f^p$ | Final population size | P | (a-e) | Fixed |
| $N_f^s$ | Final population size | S | (a-e) | Fixed |
| $\mu^p$ | Population-wide mutation rate | P | (a-c) | Inferred |
| $\mu^s$ | Population-wide mutation rate | S | (a-c) | Inferred |
| $\rho^p$ | Differential mutant fitness | P | (a) | Fixed = 1 |
| | | | (c) | Inferred |
| $\rho^s$ | Differential mutant fitness | S | (a) | Fixed = 1 |
| | | | (c) | Inferred |
| $\rho^p = \rho^s$ | Differential mutant fitness | P+S | (b) | Jointly inferred |
| $\epsilon^s$ | Extinction probability mutants | S | (a-c) | Fixed = 0 |
| $\mu_{off}$ | Mutation rate response-*off* cells | P+S | (d-e) | Jointly inferred |
| $\mathcal{S}$ | Mutation-supply ratio | S | (d-e) | Inferred |
| $f_{on}$ | Fraction response-*on* cells | S | (d) | Fixed |
| | | | (c) | Inferred if $r_{on} \neq 0$ |
| $r_{on}$ | Relative fitness response-*on* cells | S | (d) | Fixed (= 0) |
| | | | (e) | Inferred |
| $\mu_{on}$ | Mutation rate response-*on* cells | S | (d-e) | Calculated* |
| $\epsilon_{off}^s, \epsilon_{on}$ | Extinction probabilities mutants | S | (d-e) | Fixed = 0 |

The different inference models are homogeneous-response (a) without, (b) with constrained or (c) with unconstrained differential mutant fitness, and heterogeneous-response with (d) zero or (e) non-zero division rate of response-*on* cells. In the inference, each parameter is either set to a fixed value or inferred, with the exception of the mutation rate of response-*on* cells, which is calculated from $\mathcal{S}$, $\mu_{off}$ and $f_{on}$. Parameters which appear in both permissive (P) and stressful (S) conditions are jointly inferred.

$x^s$ under permissive and stressful conditions, respectively, for a given model with parameters $\theta$. Here, $m^p$ and $m^s$ represent the maximal observed numbers of mutant colonies, and $x^p(i)$ and $x^s(i)$ are the number of plates with $i$ mutant colonies under permissive and stressful conditions, respectively. The $p_i^p$ and $p_i^s$ give the probabilities to observe $i$ mutant colonies under permissive and stressful conditions, respectively, calculated from the PGFs of the mutant count distributions using recursive formulas described in [34]. Then, we use the default optimisation algorithm implemented in the Julia [36] package Optim.jl (https://julianlsolvers.github.io/Optim.jl/stable/), to find the parameters that maximise this log-likelihood function. The parameters that are estimated depend on the specific model that is considered, as described below and summarised in Table 1.

The complete documentation of all inference algorithms can be found in the file called **inference.jl** at https://github.com/LucyL-J/Quantifying-SIM.

**Parameters in the inference.**

**Homogeneous-response model.** In the homogeneous-response model, the mutant count distributions (Eqs 5 and 6) depend on the per-division mutation rates, final population sizes, and, optionally, differential mutant fitness under permissive and stressful conditions, as well as the extinction probability of mutants under stress. All parameters must either be set as inference parameters, or set to a fixed value, which could be the default value or as measured in a separate experiment. In our implementation, the per-division mutation rates, $\mu^p$ and $\mu^s$, are inferred to calculate the increase in mutation rate associated with the stress, that is, $\frac{\mu^s}{\mu^p}$. The final population sizes under permissive and stressful conditions, $N_f^p$ and $N_f^s$, are set to fixed

values, as they would typically be measured through plating a few cultures on non-selective media and colony counting. Moreover, we set the extinction probability of mutants under stress, $\epsilon^s$, to zero because we neglect cell death in the inference, which is in common with most existing approaches, but see [18].

For the differential fitness of mutants compared to non-mutants, $\rho^p$ and $\rho^s$, we consider three cases corresponding to different versions of the homogeneous-response model: (a) mutants have the same fitness as non-mutants, $\rho^p = \rho^s = 1$; (c) mutants have a different fitness than non-mutants (*unconstrained*) and two separate values, $\rho^p$ and $\rho^s$, are inferred; or (b) mutants have a different fitness than non-mutants, but the effect is constrained to be equal under permissive and stressful conditions, $\rho^p = \rho^s$. For the models (a) and (c), the mutant count distributions under permissive and stressful conditions have no joint parameters and can, therefore, be considered separately by using existing estimation methods: (a) corresponds to the standard model and (c) to the standard model with differential mutant fitness (implemented, for example, in [37]). Studies to date have followed such an approach to estimate the increase in mutation rate. Model (b), on the contrary (with constrained differential mutant fitness, arguably a reasonable null model), represents a new version of the homogeneous-response model, which is first implemented here. In this case, we estimate the model parameters by jointly maximising the log-likelihood function given in Eq 29. In the main Results, we consider all three homogeneous-response models (a-c); in section M in S1 File, we repeat the analysis for constrained mutant fitness only, i.e. models (a-b).

**Heterogeneous-response model.** In the heterogeneous-response model, the mutant count distributions under permissive (Eq 7) and stressful conditions (Eqs 23 and 24), depend on the per-division mutation rates of response-*off* and response-*on* cells, the extinction probabilities of response-*off* and response-*on* mutants under stress, the fraction and relative fitness of response-*on* compared to response-*off* cells, and the total final population sizes under permissive and stressful conditions. The mutation rate of response-*off* cells, $\mu_{off}$, appears as a parameter in both the mutant count distributions under permissive and under stressful conditions, and the joint inference crucially relies on our assumption that, even though the division rate itself might change under stress, the *per-division* mutation rate of response-*off* cells is the same under both conditions. As for the homogeneous-response model, we assume that the final population sizes, $N_f^p$ and $N_f^s$, are known from plating on non-selective medium. Moreover, we neglect cell death, which implies $\epsilon_{off}^s = \epsilon_{on} = 0$, but test the robustness of this assumption.

For the relative fitness of response-*on* cells, we consider two different model versions of the approximate heterogeneous-response model: (d) as a default, we set $r_{on} = 0$, inspired by the SOS response in *E. coli*, which inhibits cell division, or to a small non-zero value that we assume is measured in a separate experiment (section H in S1 File); and (e) we infer $r_{on}$. Similarly, for the fraction of the response-*on* subpopulation, we (i) assume that it is a known quantity measured in a separate experiment (e.g. microscopy or flow cytometry), or (ii) set it as an inference parameter.

Ultimately, we are interested in quantifying the relative increase in mutation rate associated with induction of the stress response, that is, $\frac{\mu_{on}}{\mu_{off}}$. To do so we need to estimate the mutation rate of response-*on* cells, $\mu_{on}$. However, we do not directly infer this parameter; instead, we infer the composite parameter of the mutation-supply ratio $\mathcal{S}$ defined in Eq 27, from which we calculate $\mu_{on}$. The reason for this approach is that for $r_{on} = 0$, the mutant count distribution under stress does not depend separately on $\mu_{on}$ and $f_{on}$ but only on $\mathcal{S}$ (together with $\mu_{off}$ and $N_f^s$); see Eq 26. This also implies that, for $r_{on} = 0$ and when the fraction of the response-*on*

subpopulation is unknown, $\mu_{on}$ and thus $\frac{\mu_{on}}{\mu_{off}}$ cannot be calculated. In this case, we report estimates of $\mathcal{S}$ instead as an indicator of heterogeneity.

**Confidence intervals using profile likelihood.** In addition to the maximum-likelihood estimates, we calculate 95% confidence intervals using a profile likelihood approach (section E in S1 File). The confidence interval for each parameter contains all values such that, after optimisation over the remaining inference parameters, the likelihood does not significantly drop according to a likelihood ratio test.

## Evaluating inference methods on simulated data

We test our estimation method using simulated fluctuation assay data: For chosen ranges of parameter values, we perform stochastic simulations of the population dynamics during the growth phases of a pair of fluctuation assays, one under permissive and the other under stressful conditions. From the resulting mutant count distributions, we infer the respective parameters under heterogeneous and homogeneous-response models and compare the estimated with the true simulated parameters, as well as perform model selection. For most of this study, we simulate under the heterogeneous-response model, but we repeat part of the analysis for simulation under the homogeneous-response model (sections K and N in S1 File).

The complete documentation of all population dynamic functions can be found in the file called **population_dynamics.jl** at https://github.com/LucyL-J/Quantifying-SIM.

**Simulation methods.** To simulate the growth phase under permissive conditions, we consider exponential growth of the non-mutant population (Eq 1) and implement the occurrence of mutations as a time-inhomogeneous Poisson process with rate proportional to the population size using a standard approach described, for example, in [38]. We treat mutant cells stochastically by using Gillespie's algorithm [39] to simulate the pure birth process described by Eq 2 with zero death rate. In the case of the homogeneous-response model, the population growth rate is given by the division rate, $\gamma^p$, the mutation rate per cell per unit time by $\nu^p$ and the birth rate of mutants by $\rho^p \cdot \gamma^p$. Similarly, for the heterogeneous-response model, the rates are given by the respective rates of response-*off* cells ($\gamma_{off}^p$, $\nu_{off}^p$ and $\rho_{off}^p \cdot \gamma_{off}^p$).

For the homogeneous-response model, we simulate the growth phase under stressful conditions using the same algorithm but with different rates ($\gamma^s$, $\nu^s$, $\rho^s \cdot \gamma^s$). To simulate stressful conditions under the heterogeneous-response model, we use Eqs 10 and 11 (setting $n_{on}(0) = 0$) to describe the growth of the response-*off* and response-*on* subpopulations, and implement the occurrence of mutations as two independent time-inhomogeneous Poisson processes with rates proportional to the subpopulation sizes ($n_{off}^s(t)$ and $n_{on}(t)$, respectively) and the mutation rates per cell per unit time ($\nu_{off}^s$ and $\nu_{on}$). We simulate the mutant dynamics stochastically as a two-type branching process described by Eq 12 using Gillespie's algorithm.

In all simulations, we set the duration $t_f$ of the growth phase such that the expected number of *mutations* (not *mutants*) is constant across the considered parameter ranges (section C in S1 File). In our main results, we take $c = 50$ parallel cultures per assay, which is readily feasible if culturing on a 96-well plate; see, for example, [16] for a protocol. In sections D and N in S1 File, we also examine the sensitivity of the results to the number of parallel cultures by considering smaller numbers $c$.

**Accuracy and precision of parameter estimates.** Generally, we evaluate the accuracy and precision of all estimation methods by simulating pairs of fluctuation assays, estimating the parameters of the inference model and comparing the respective estimates with the true values; repeated $R = 100$ times for each parameter set simulated. We consider the deviation from the true value of the median estimate across the simulations as a measure of the accuracy of the estimation and the variation as a measure of the precision. In particular, we calculate the

median of the relative error across the $R$ replicates,

$$\text{RE} = \frac{\text{median}\left(\left\{\frac{\mu_{on}}{\mu_{off}}\right\}_{\text{est}} - \left(\frac{\mu_{on}}{\mu_{off}}\right)_{\text{true}}\right)}{\left(\frac{\mu_{on}}{\mu_{off}}\right)_{\text{true}}}. \tag{30}$$

Here, a positive or negative relative error implies over- or underestimation, respectively. Moreover, we calculate the coefficient of variation across the $R$ replicates,

$$\text{CV} = \frac{\text{std}\left(\left\{\frac{\mu_{on}}{\mu_{off}}\right\}_{\text{est}}\right)}{\text{mean}\left(\left\{\frac{\mu_{on}}{\mu_{off}}\right\}_{\text{est}}\right)}. \tag{31}$$

where 'std' denotes standard deviation. Where we calculate confidence intervals (section E in S1 File), we also use the median width of the confidence intervals as a measure of precision.

To plot the estimated parameters across the $R = 100$ simulations, we use boxplots, where each box shows the median and interquartile range with whiskers extending to 1.5 times the interquartile range and any outliers outside that range represented as dots. To summarise the confidence intervals on each of the $R = 100$ estimates, we plot the median maximum likelihood estimate with error bars extending to the medians of the lower and upper bounds of the 95% confidence intervals.

**Model selection: Heterogeneous versus homogeneous response.** We also evaluate whether it is possible to identify the heterogeneity of stress responses from mutant count data alone. In this case, we suppose we do not have separate experimental data showing heterogeneity and, therefore, do not have an estimate of $f_{on}$. For this purpose, we simulate fluctuation assays under the (exact) heterogeneous-response model and under the homogeneous-response model (sections K and N in S1 File) and, then, fit the different homogeneous (a-c) and (approximate) heterogeneous-response models (d-e). For model selection, we use a combination of likelihood-ratio test (LRT) and Akaike information criterion (AIC). The AIC is defined as

$$\text{AIC} = 2(k - \ln \mathcal{L}) \tag{32}$$

where $k$ is the number of inferred parameters of the model. Within the set of models (a-c) and within the set (d-e), the models are nested and we use LRTs to determine the best heterogeneous/homogeneous-response model within each set first. However, we cannot use LRT to select between the sets (a-c) and (d-e) because these models are not nested. Therefore, between the two best models, we select the one with the lowest AIC. However, if the difference in AIC is within ±2, we say that the AICs are comparable, and neither of the models can be selected. We also consider the Bayesian information criterion (BIC) as an alternative second selection step (section N in S1 File).

## Results

We aim to estimate the increase in mutation rate associated with the induction of the stress response when this response is heterogeneously expressed across the bacterial population. In particular, we consider cases in which the population can be divided into two discrete subpopulations: one with a low expression level (response *off*) and the other with a high expression level (response *on*). The key principle of our method is to jointly infer their mutation rates from mutant count data obtained from a pair of fluctuation assays under permissive and stressful conditions. For the latter, we need to disentangle the contributions from the

response-*off* and response-*on* subpopulations. The success of this method relies on the changing shape of the mutant count distribution under stress, which occurs if there is a highly mutating but slowly dividing response-*on* subpopulation (Fig B in S1 File).

To evaluate the performance of our method, we use simulated mutant count data to compare the estimated parameters with the true simulated ones. First, we explore how the accuracy and precision of our method depend on the model parameters by simulating and inferring under the same model. Then, we test the robustness of our method to model deviations by simulating under a more complex model than used in the inference. Finally, we determine under what conditions the heterogeneous-response model can be distinguished from the homogeneous-response model assumed in currently available methods by inferring under both models and comparing how well they fit simulated data.

In all simulations, we set the initial population size to $10^4$ and the initial fraction of the response-*on* subpopulation to zero. Moreover, we consider the duration of the growth phase such that the expected number of mutations equals one. This way, the resulting number of resistant mutant colonies on each selective plate is usually within an experimentally countable range of zero to a couple hundred (section C in S1 File). Table 2 summarises the default parameters used in the simulations, while parameters that vary are specified in the relevant Results section. For each parameter combination, we simulate $R = 100$ pairs of fluctuation assays under permissive and stressful conditions, with $c = 50$ parallel cultures per assay. We also test the sensitivity of our method's performance to the number of parallel cultures (section D in S1 File) and repeat the model selection analysis for smaller numbers of parallel cultures ($c = 20, 10$) in section N in S1 File. Generally, we assume that the final population sizes in permissive and stressful conditions ($N_f^p$ and $N_f^s$, respectively) and the fraction of the response-*on* subpopulation under stress ($f_{on}$) are known from separate experimental measurements, except for the last Results section where we infer without a separate estimate of $f_{on}$.

## Estimation of the mutation-rate increase is accurate and precise for sufficiently large response-*on* mutation supply

First, we evaluate our novel inference method's performance in the best-case scenario; that is, we simulate and infer under the same model: a model of heterogeneous stress responses without cell death, with mutant fitness equal to non-mutant fitness, and with zero division rate of response-*on* cells. We simulate for a range of mutation rates in response-*on* cells, $\nu_{on} \in [10^{-5}, 10^{-8}]\ h^{-1}$ and switching rates $\alpha \in [0.001, 0.1]\ h^{-1}$. Note that the per-division mutation rate $\mu_{on}$ and the per-unit-time rate $\nu_{on}$ are equivalent here because we set the division rate $\gamma_{off}^s$ to one.

For the inference, we consider the same model as used in the simulations with the only exception that we neglect switching *on* the stress response in mutants and initial population size effects; for a comparison of the exact and approximated heterogeneous-response model, see Model and methods. For each set of mutant count data, we infer the mutation rate of response-*off* cells ($\mu_{off}$) and the mutation-supply ratio ($\mathcal{S}$), which defines the relative contribution of response-*on* cells to the total mutation supply (Eq 27). From these estimates, we calculate the stress-induced mutation-rate increase, i.e. $\frac{\mu_{on}}{\mu_{off}}$. To quantify the accuracy of our method, we calculate the median of the relative error of our estimated mutation-rate increases (Eq 30). Additionally, we use the coefficient of variation across the estimates (Eq 31) to quantify our method's precision.

Comparing the estimated with the true mutation-rate increase $\frac{\mu_{on}}{\mu_{off}}$, we find that the accuracy and precision improve with increasing $\frac{\mu_{on}}{\mu_{off}}$ and relative switching rate $\tilde{\alpha}$ (Fig 2B and 2C). For example, for $\tilde{\alpha} = 0.05$, when $\frac{\mu_{on}}{\mu_{off}} \geq 25$, 95% of estimates lie within 2–fold of the true mutation-rate increase and the estimation is unbiased (Fig 2A). For smaller $\frac{\mu_{on}}{\mu_{off}}$, on the other hand, the

**Table 2. Parameters used in the simulations unless explicitly specified otherwise.**

| Symbol | Cond. | Definition | Numerical value |
|---|---|---|---|
| $\gamma^p_{off}$ | P | Division rate response-*off* cells | $1\ h^{-1}$ |
| $\rho^p_{off}$ | P | Differential mutant fitness response-*off* cells | 1 |
| $\nu^p_{off}$ | P | Mutations per unit time response-*off* cells | $10^{-8}\ h^{-1}$ |
| $\gamma^s_{off}$ | S | Division rate response-*off* cells | $1\ h^{-1}$ |
| $\rho^s_{off}$ | S | Differential mutant fitness response-*off* cells | 1 |
| $\nu^s_{off}$ | S | Mutations per unit time response-*off* cells | $10^{-8}\ h^{-1}$ |
| $\delta^s_{off}$ | S | Death rate response-*off* cells | $0\ h^{-1}$ |
| $\gamma_{on}$ | S | Division rate response-*on* cells | $0\ h^{-1}$ |
| $\rho_{on}$ | S | Differential mutant fitness response-*on* cells | 1 |
| $\nu_{on}$ | S | Mutations per unit time response-*on* cells | $10^{-6}\ h^{-1}$ |
| $\delta_{on}$ | S | Death rate response-*on* cells | $0\ h^{-1}$ |
| $\alpha$ | S | Switching rate | $0.05\ h^{-1}$ |

For simplicity, we set the division rate of response-*off* cells under permissive (P) and stressful (S) conditions to $1\ h^{-1}$. The switching rate for the SOS response in *E. coli* has been estimated using single-cell imaging [27]. The mutation rate is based on rifampicin resistance, a selective marker commonly used in fluctuation assays. In *E. coli*, the number of point mutations conferring rifampicin resistance has been estimated to be 79 [40] and the mutation rate between $0.2 \cdot 10^{-10}$ and $5 \cdot 10^{-10}$ nucleotides per generation in permissive conditions [41], yielding a mutation rate of $\nu^p_{off}, \nu^s_{off} \approx 10^{-8}$ per unit time for response-*off* cells. Meanwhile, the mutation rate under induction of the stress response ($\nu_{on}$) is set to a default of 100 times higher, comparable to the increase associated with genetic mutators [41, 42].

variation in the estimates becomes large. Nonetheless, if the mutation-rate increase is estimated to be $>25$, the true increase is very likely to be $>10$, and conversely, if the mutation-rate increase is estimated close to zero ($<10^{-3}$), it is very likely to be $<10$.

The mutation-supply ratio, which is defined $\mathcal{S} := \frac{\mu_{on}}{\mu_{off}} \frac{f_{on}}{1-f_{on}}$ (approximately given by the product of $\frac{\mu_{on}}{\mu_{off}}$ and $\tilde{\alpha}$ for small $\alpha$ and small $r_{on}$) determines our method's performance. For $\mathcal{S} \gg 1$, the contribution of the response-*on* subpopulation is dominating. In contrast, for $\mathcal{S} \ll 1$, the response-*on* subpopulation contributes very little to the total mutant count. Overall, in the best-case scenario and for the parameter regime considered here, $\mathcal{S} \sim \mathcal{O}(1)$ or greater is sufficient for an accurate and precise estimate of the mutation-rate increase.

We also evaluate the sensitivity of our method's performance to the number of parallel cultures (Fig C in S1 File). For smaller $c$, the precision of our method worsens compared to $c = 50$, but the estimation of $\frac{\mu_{on}}{\mu_{off}}$ remains accurate for sufficiently large $\mathcal{S} \sim \mathcal{O}(1)$ or greater.

In addition to the maximum likelihood estimates, we calculate 95% profile likelihood confidence intervals on the estimates of the mutation-rate increase (Fig D in S1 File). We find that the median width of the confidence intervals increases with decreasing $\frac{\mu_{on}}{\mu_{off}}$ and $\tilde{\alpha}$, in a similar way as the CV of the estimates for $R = 100$ repeated simulations shown in Fig 2C. Moreover, we confirm that the true value for the mutation-rate increase lies outside of the confidence interval in $< 5\%$ of the simulations.

## Cell death has a limited impact on estimates

Our inference model accounts for changes in mutation and division rates upon induction of the stress response but neglects other potential consequences of the stress, such as cell death.

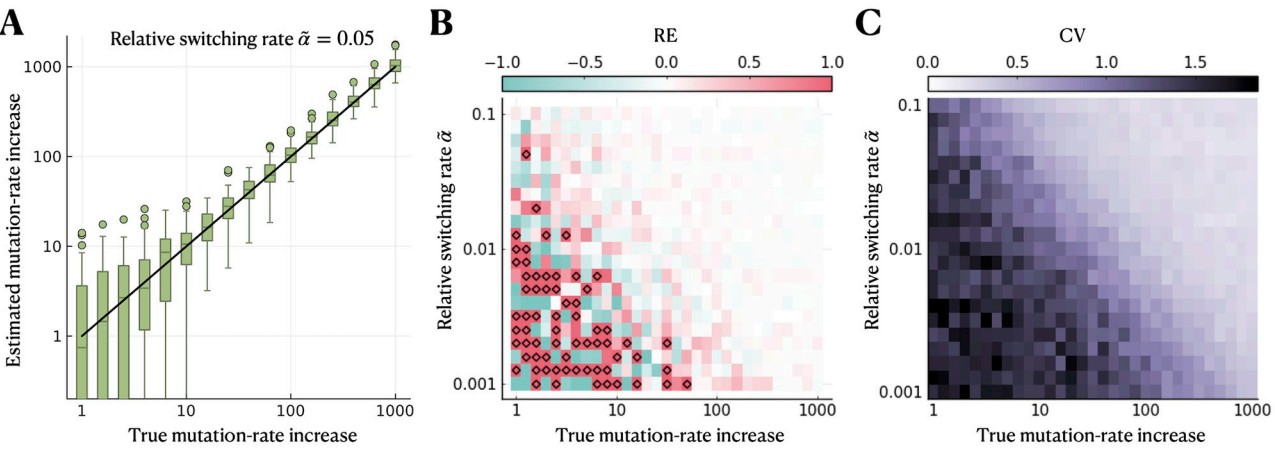

**Fig 2. Estimation of the mutation-rate increase is accurate and precise when the mutation-supply ratio is sufficiently large.** We simulate using the simplest model of heterogeneous stress responses (without cell death or differential mutant fitness and with zero division rate of response-*on* cells) and infer the mutation-rate increase $\frac{\mu_{on}}{\mu_{off}}$ assuming the same model in the inference. **A** Estimated compared to true $\frac{\mu_{on}}{\mu_{off}}$ for a range of values of $\frac{\mu_{on}}{\mu_{off}}$ and a relative switching rate of $\tilde{\alpha} = 0.05$. **B** Median relative error of estimated compared to true mutation-rate increase for a range of $\frac{\mu_{on}}{\mu_{off}}$ and $\tilde{\alpha}$. Over/underestimation is shown in red/blue, and diamonds indicate a relative error greater than one. **C** Coefficient of variation across estimates. The parameter ranges used in the simulations are $\nu_{on} \in [10^{-5}, 10^{-8}] \ h^{-1}$ and $\alpha \in [0.001, 0.1] \ h^{-1}$.

Previous work showed that the occurrence of cell death, if neglected in the inference model, leads to overestimation of mutation rate [18]. Therefore, we asked whether neglecting cell death has a similar effect in the case of heterogeneous stress responses. For this purpose, we simulate fluctuation assays under an extended model of heterogeneous stress responses with cell death. We consider the cases that (i) only response-*off* cells are affected by cell death, (ii) only response-*on* cells are affected, or (iii) all cells are affected equally, using parameter ranges of $\delta_{off}^s$, $\delta_{on} \in [0.0, \ 0.5] \ h^{-1}$.

Interestingly, we find that any biases in the estimated mutation-rate increase depend on which subpopulation is affected by cell death. If only response-*off* cells die, the mutation-rate increase is overestimated for sufficiently large death rates (Fig 3A). On the other hand, if only response-*on* cells die, the mutation-rate increase is underestimated (Fig 3B). The estimation remains unbiased if both subpopulations are equally affected by cell death. However, the variation in the estimates increases for large death rates (Fig 3C). From the contribution of the response-*on* subpopulation to the mutant count given in Eq 24, it can be seen that the effects of cell death in response-*off* and response-*on* cells partly cancel each other out. This result also holds for a smaller switching rate (Fig E in S1 File).

We test another biologically realistic model deviation: the fitness of mutants differs from non-mutants during the growth phase (Fig F in S1 File). We find that neglecting this effect in the inference causes very little bias in the estimates for either a fitness advantage or a fitness cost of mutations.

## Estimation remains accurate when response-*on* cells have a small to moderate division rate

So far, we considered response-*on* cells not to divide at all during the growth phase, motivated by the SOS response. However, the division rate of response-*on* cells might be non-zero, especially if cells are exposed to a very low level of DNA damage (in the case of SOS) or for other stress responses. As a default setting, our inference method sets the relative fitness of response-

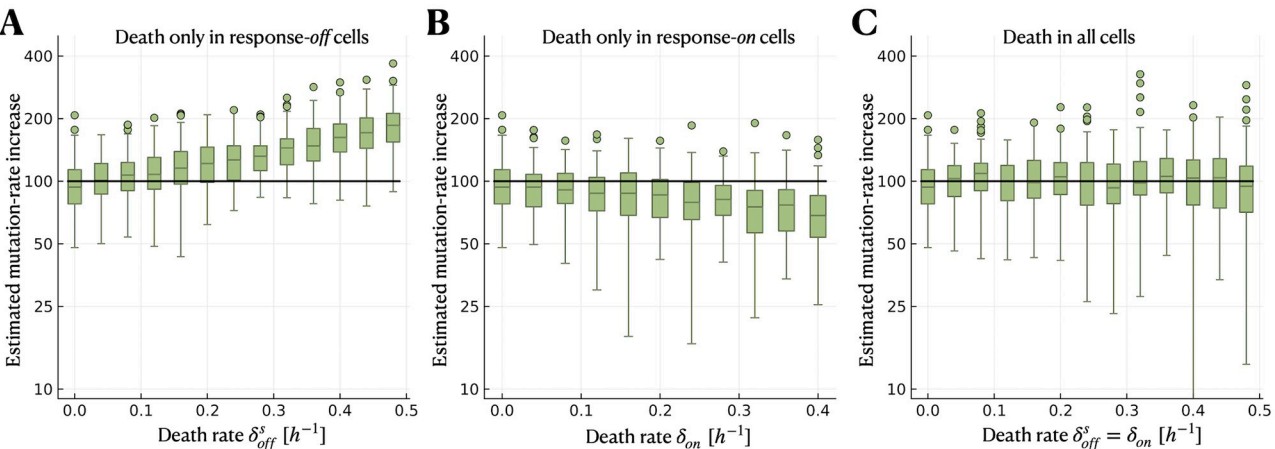

**Fig 3. Cell death has limited impact on the estimation of the mutation-rate increase.** We simulate using the heterogeneous-response model (without differential mutant fitness and with zero division rate of response-*on* cells) but with cell death. However, we neglect cell death in the model used for inference. The black solid lines indicate the true mutation-rate increase used in the simulations. **A** Estimated mutation-rate increase when only response-*off* cells are affected by cell death, **B** when only response-*on* cells are affected by cell death, and **C** when all cells are affected by cell death equally. The parameter ranges used in the simulations are $\delta^s_{off}$, $\delta_{on} \in [0.0,\ 0.5]\ h^{-1}$ (with $\gamma^s_{off} = 1\ h^{-1}$).

*on* cells to zero ($r_{on} = 0$), but it also allows us to estimate $r_{on}$ as an inference parameter. In the following, we evaluate the performance of our inference method when true $r_{on} > 0$, specifically the impact on the estimated mutation-rate increase.

We simulate under the heterogeneous-response model with a non-zero division rate of response-*on* cells, considering a parameter range of $\gamma_{on} \in [0, 1]\ h^{-1}$ (with $\gamma_{off} = 1\ h^{-1}$). Note that the relative fitness of response-*on* cells $r_{on}$ is equivalent to their division rate $\gamma_{on}$ because we consider no cell death here. We consider two different inference approaches. In one case, we infer the mutation rate $\mu_{off}$, the mutation-supply ratio $\mathcal{S}$ and the relative fitness of response-*on* cells $r_{on}$. Alternatively, we infer only $\mu_{off}$ and $\mathcal{S}$ while setting $r_{on} = 0$. We estimate the mutation-rate increase $\frac{\mu_{on}}{\mu_{off}}$ in both cases and compare it to the true value. In the first case, we also compare the estimated with the true $r_{on}$.

We find that the estimation of the mutation-rate increase remains accurate for small to moderate relative fitness $r_{on}$. For larger $r_{on} \to 1$, on the other hand, the mutation-rate increase is underestimated, yet more accurate and precise if $r_{on}$ is also inferred (Fig 4A). The estimate of $r_{on}$ itself is also underestimated for larger $r_{on}$ (Fig 4B).

We also evaluate the performance when $r_{on}$ is set to the true value in the inference. Interestingly, this increases the accuracy and precision of the estimate of $\frac{\mu_{on}}{\mu_{off}}$ only slightly compared to when $r_{on}$ is inferred (Fig G in S1 File). The reason lies in the approximation made to derive the mutant count distribution (Eq 13), which is no longer valid as $r_{on} \to 1$.

## Model selection between heterogeneous and homogeneous response is often inconclusive

In many cases, it may not be known *a priori* whether the stress response is heterogeneously expressed across the population or whether, in contrast, all cells respond similarly. We want to determine whether distinguishing these two models is possible using mutant count data from fluctuation assays alone. To do so, we simulate fluctuation assays under the heterogeneous-response model for a range of relative fitness of response-*on* cells, $r_{on}$. For the inference, we use

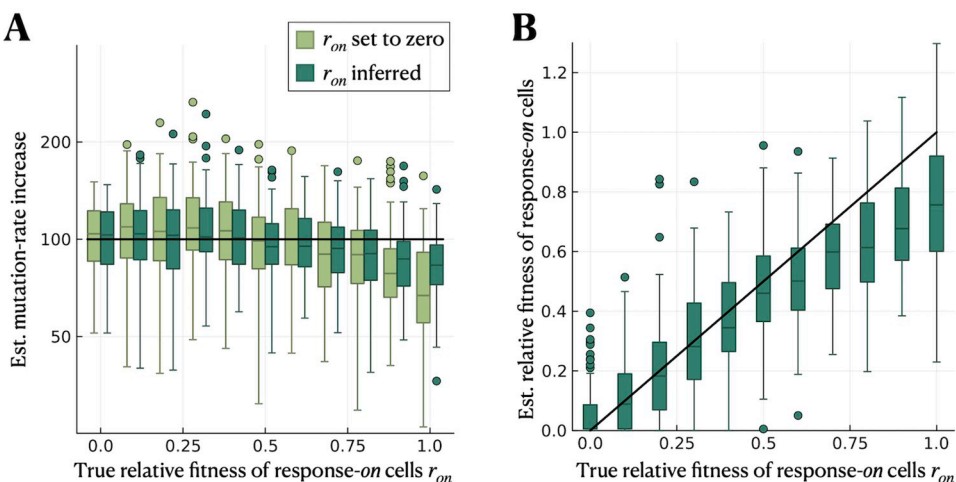

**Fig 4. Estimation of the mutation-rate increase remains accurate when response-*on* cells have a small to moderate relative fitness.** We simulate using the heterogeneous-response model (without cell death or differential mutant fitness) with $r_{on} \geq 0$ being the relative fitness of response-*on* cells compared to response-*off* cells. We consider two cases for the inference: (i) setting $r_{on}$ to zero and only inferring $\mu_{off}$ and $\mathcal{S}$, and (ii) inferring $r_{on}$ in addition. **A** Estimated mutation-rate increase $\frac{\mu_{on}}{\mu_{off}}$ for both cases and a range of relative fitness of response-*on* cells. The solid black line indicates the true value of $\frac{\mu_{on}}{\mu_{off}}$. **B** Estimated compared to true relative fitness of response-*on* cells in inference case (ii). The parameter range used in the simulations is $\gamma_{on} \in [0, 1]$ in $h^{-1}$.

both the heterogeneous- and homogeneous-response models and compare how well they fit the data. We use the same simulation data as in the previous section (parameter range $\gamma_{on} \in [0, 1]$ $h^{-1}$). However, we suppose that the fraction of the response-*on* subpopulation $f_{on}$ is unknown. Therefore, when using the heterogeneous-response model in the inference, we either set $r_{on} = 0$ (in which case $f_{on}$ drops out of the equations) and infer only $\mu_{off}$ and $\mathcal{S}$, or set $f_{on}$ and $r_{on}$ as additional inference parameters. Note that, if $f_{on}$ is not inferred, the mutation-rate increase $\frac{\mu_{on}}{\mu_{off}}$ can no longer be calculated, see Eq 26.

We perform model selection between the heterogeneous and the homogeneous-response models using a combination of the likelihood ratio test (LRT) and the Akaike Information Criterion (AIC), which consider how well the models reproduce the data while penalising the number of model parameters. For the homogeneous-response model, we consider three cases: (a) without differential mutant fitness (inference parameters: $\mu^p$ and $\mu^s$), (b) with differential mutant fitness, but constrained to be equal under permissive and stressful conditions (inference parameters $\mu^p$, $\mu^s$ and $\rho^p = \rho^s$) and (c) with unconstrained differential mutant fitness (inference parameters: $\mu^p$, $\mu^s$, $\rho^p$ and $\rho^s$). For the heterogeneous-response model, we consider two cases: (d) zero relative fitness of response-*on* cells (inference parameters: $\mu_{off}$ and $\mathcal{S}$) and (e) non-zero relative fitness of response-*on* cells (inference parameters: $\mu_{off}$, $\mathcal{S}$, $f_{on}$ and $r_{on}$). We use LRTs to select the best homogeneous model (a-c) and the best heterogeneous model (d-e) within each of these sets of nested models. Then, we use the AIC to select between the best homogeneous and the best heterogeneous-response model, which are not nested. If the difference in AIC is less than two, we say neither model is clearly selected.

After applying this two-step model selection, we find that the heterogeneous-response model is selected in the majority of cases (around 75% for $r_{on} = 0$) when the relative fitness of response-*on* cells is small (Fig 5A). For increasing $r_{on}$, however, the homogeneous-response model without differential mutant fitness is selected with increasing frequency until it is selected for the majority of simulations for large $r_{on}$. The other models are selected for only a

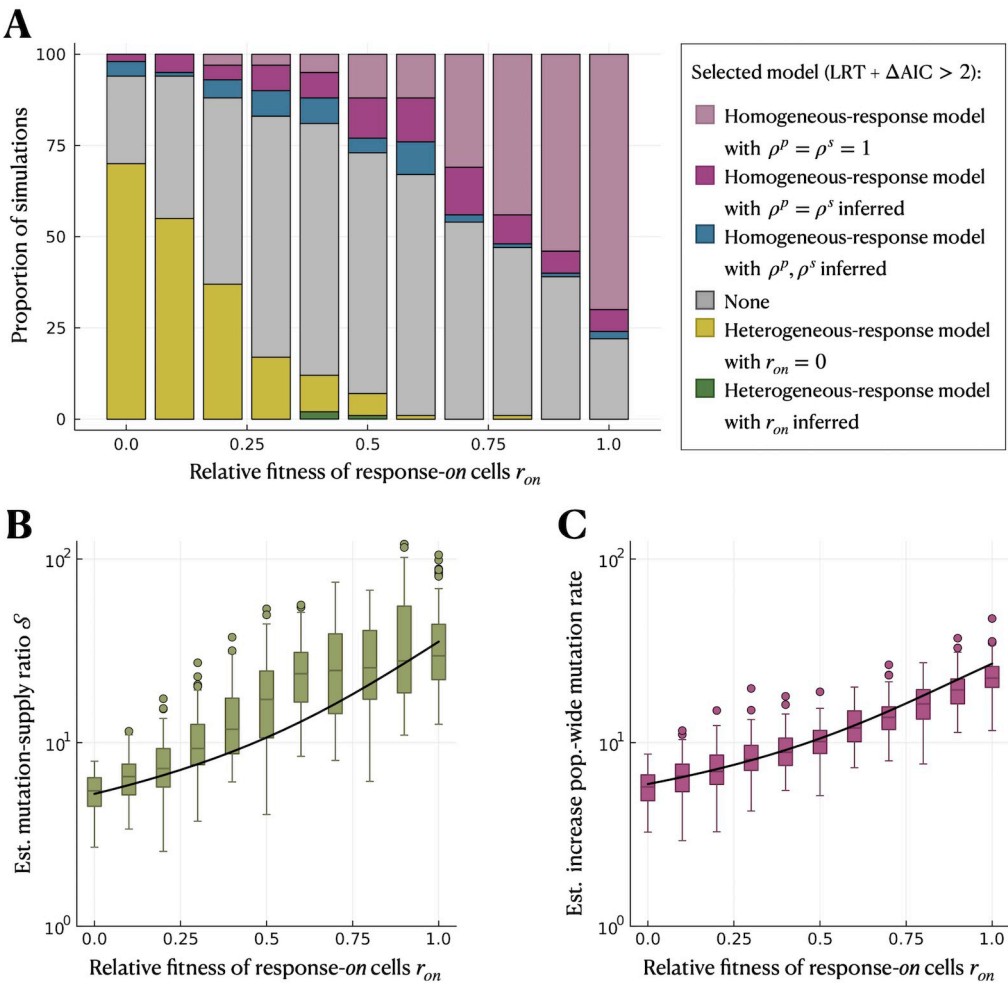

**Fig 5. The heterogeneous-response model is selected only when response-*on* cells have a small relative fitness.** We simulate under the heterogeneous-response model for a range of relative fitness of response-*on* cells, $r_{on}$. In the inference, we use (d) the heterogeneous-response model with $r_{on} = 0$ (yellow) and (e) the heterogeneous-response model with $r_{on}$ and $f_{on}$ as inference parameters (dark green) to infer the mutation rate of response-*off* cells, $\mu_{off}$, and mutation-supply ratio, $\mathcal{S}$. We also use the homogeneous-response model (a) without differential mutant fitness ($\rho^p = \rho^s = 1$; light purple), (b) with constrained differential mutant fitness ($\rho^p = \rho^s$ inferred; dark purple) and (c) with unconstrained differential mutant fitness ($\rho^p, \rho^s$ inferred; blue) to infer the population-wide mutation rates under permissive and stressful conditions, $\mu^p$ and $\mu^s$, and, for (b) and (c), additionally $\rho^p$ and $\rho^s$. **A** Model selection using LRT and AIC. **B** Estimated mutation-supply ratio, $\mathcal{S}$, by the best heterogeneous-response model. **C** Estimated increase in mutation rate, $\frac{\mu^s}{\mu^p}$, by the best homogeneous-response model. The black lines in **B** and **C** indicate the true values of $\mathcal{S}$ and the increase in population mean mutation rate, $\bar{M}$, respectively. The parameter range used in the simulations is $\gamma_{on} \in [0, 1]\ h^{-1}$.

small number of simulations. Over the whole parameter range, there is a substantial fraction of cases in which no model can be selected, with the highest proportion of $\approx$50% for intermediate $r_{on}$. This implies that heterogeneity in stress responses with sufficiently large $\mathcal{S}$ can in principle be detected, but only if the division rate of response-*on* is very small (or zero).

Comparing the mutation-supply ratio $\mathcal{S} = \frac{\mu_{on} f_{on}}{\mu_{off}\,(1-f_{on})}$ estimated by the best heterogeneous-response model with its true value, we find that the estimate is accurate and precise for small $r_{on}$, but with a slight loss in precision for larger $r_{on}$ (Fig 5B). This means that even without a

separate estimate of $f_{on}$, the magnitude of the heterogeneity in mutation rates (in the form of $\mathcal{S}$) can be captured.

We also compare the estimated increase in mutation rate $\frac{\mu^s}{\mu^p}$ from the best homogeneous-response model with the true increase in population mean mutation rate under the simulated heterogeneous-response model, given by $\bar{M} = \frac{(1-f_{on})\mu_{off}+f_{on}\mu_{on}}{\mu_{off}}$. Interestingly, the inferred $\frac{\mu^s}{\mu^p}$ is an accurate and precise estimate of $\bar{M}$ over the whole range of $r_{on}$, with only a slight underestimation for large $r_{on}$ (Fig 5C). For $r_{on} \rightarrow 1$ and assuming no cell death, accurate estimation of $\bar{M}$ is expected because the probability generating function (PGF) of the mutant count distribution reduces to

$$G_{\text{het}}^s(z) = \exp\left[\left(\mu_{off}(1-f_{on})+\mu_{on}f_{on}\right)N_f^s\frac{(1-z)\log(1-z)}{z}\right], \tag{33}$$

This distribution is equivalent to the homogeneous-response model without differential mutant fitness (Eq 6), which is selected as the best homogeneous-response model for most simulations. For small $r_{on}$, on the other hand, homogeneous-response models with differential mutant fitness are selected more often, and they (falsely) infer an increasingly severe mutant cost under stressful conditions as $r_{on} \rightarrow 0$ (Fig H in S1 File), despite mutations not having a cost in the simulations.

We repeat the analysis for a smaller mutation-rate increase, $\frac{\mu_{on}}{\mu_{off}} = 10$, and find that the heterogeneous-response model is selected less often, only in around 25% of the simulations for $r_{on} = 0$ (Fig I in S1 File), implying that small mutation-rate increases are most likely not picked up through model selection. We also perform model selection when using smaller numbers of parallel cultures ($c = 20, 10$) in the inference, and find that, overall, model selection is less informative for smaller $c$ (Fig M in S1 File).

Finally, we check for the rate of false positives where the heterogeneous-response model is incorrectly selected in the absence of heterogeneity, by simulating under versions of the homogeneous-response model and performing model selection. We find that, when simulating under the homogeneous-response model with constrained mutant fitness, the homogeneous-response model is selected in almost all cases independent of the increase in mutation rate (Fig N in S1 File). Therefore, if the mutant fitness is the same under permissive and stressful conditions, the risk of false positives is negligible. When simulating under the homogeneous-response model with unconstrained mutant fitness, on the other hand, there are more cases in which no model or the heterogeneous-response model is selected (Fig O in S1 File).

When simulating under the homogeneous-response model without an increase in mutation rate ($\frac{\mu^s}{\mu^p} = 1$) and with small mutant fitness costs, no model can be selected in most cases, but both heterogeneous and homogeneous-response models correctly infer that there is no increase in mutation rate, corresponding to $\mathcal{S} = 0$ and $\bar{M} = 1$, respectively (Fig J in S1 File).

## Discussion

Since its introduction 80 years ago, the standard model behind the fluctuation assay has been extended numerous times to overcome limitations and make it more biologically realistic. Extensions particularly relevant for quantifying stress-induced mutagenesis include considering cell death [18, 43] and differential mutant fitness [44]. In this study, we addressed a so-far overlooked limitation of fluctuation analysis: heterogeneity in the expression of stress responses, which single-cell studies have recently demonstrated. Our population dynamic model considers that only a subpopulation of cells (fraction $f_{on}$) have the stress response

switched *on* and the remainder of the population switched *off*. This allows us to estimate the relative increase in mutation rate associated with the induction of the stress response, $\frac{\mu_{on}}{\mu_{off}}$.

We tested our estimation method with simulated mutant count data, which confirmed accurate and precise estimation of $\frac{\mu_{on}}{\mu_{off}}$ for sufficiently large mutation-supply ratio defined as $\mathcal{S} := \frac{f_{on}}{1-f_{on}} \frac{\mu_{on}}{\mu_{off}}$ (Fig 2). $\mathcal{S}$ depends on the mutation-rate increase itself and the fraction of the response-*on* subpopulation. While $\frac{\mu_{on}}{\mu_{off}}$ is inherent to the stress response, $f_{on}$ could potentially be increased through experimental design. For example, increasing the antibiotic concentration has been shown to increase the rate of switching *on* the stress response and thus the fraction of the response-*on* subpopulation [27]. Our results suggest that mutation rate estimates would be more accurate at higher antibiotic concentrations, all else being equal. Increasing antibiotic concentration could, however, also increase cell death. We neglect cell death in our inference, but we showed that our method is robust to this model deviation up to moderate death rates when cell death affects response-*off* and response-*on* subpopulations equally (Fig 3C).

We used model selection with a combination of likelihood-ratio tests and AIC to evaluate whether a signal of mutation-rate heterogeneity can be detected from fluctuation assays alone. The chances of detecting heterogeneity are highest when response-*on* cells are non- or only slowly-dividing ($r_{on} \ll 1$). For moderate switching rates and a mutation-rate increase of $\frac{\mu_{on}}{\mu_{off}} = 100$, the heterogeneous-response model is selected over homogeneous-response models in the majority of the simulated experiments (Fig 5A). However, with increasing $r_{on}$ ($> 0.25$), the heterogeneous-response model is only rarely selected. A smaller mutation-rate increase $\frac{\mu_{on}}{\mu_{off}} = 10$ also cannot effectively be detected by model selection (Fig I in S1 File). Generally, model selection with fewer than $c \approx 50$ parallel cultures per fluctuation assay will be very difficult to interpret even for the best-case parameter range (Fig M in S1 File).

Our results suggest that heterogeneity in stress responses may have been overlooked when using fluctuation assays, and these data should be complemented with additional experiments to support or rule out alternative explanations. For example, mutants arising in the fluctuation assay can be isolated and their relative fitness compared to non-mutants measured with a pair of competitive fitness assays under permissive and stressful conditions. This measurement would allow researchers to check whether mutant fitness values estimated from the homogeneous-response model fit ($\rho^p$ and $\rho^s$) are reasonable. In particular, a large difference in estimated $\rho^p$ and $\rho^s$ may alternatively indicate the presence of a slowly-dividing and highly-mutating subpopulation (Fig H in S1 File). Constraining $\rho^p = \rho^s$, arguably a reasonable null model, increases the fraction of simulated experiments in which the heterogeneous model is selected (Fig L in S1 File).

If there is reason to suspect heterogeneity in the stress response, experimentalists can test this hypothesis directly by engineering fluorescent reporters into the bacterial strain of interest and measuring the response on a single-cell level, e.g. by flow cytometry [8, 9, 24] or microscopy [8, 19, 20]. These experiments would provide an independent estimate of the fraction of the response-*on* subpopulation to further constrain the heterogeneous-response model and allow calculation of $\frac{\mu_{on}}{\mu_{off}}$. In reality, multiple factors causing deviation from the standard fluctuation assay model (e.g. heterogeneous stress responses, differential mutant fitness, and cell death) will likely operate simultaneously. Since it is not feasible to reliably estimate a large number of parameters from fluctuation assay data alone, separate experiments become important to decide which deviation(s) are most relevant to incorporate into the fluctuation analysis.

Interestingly, the homogeneous-response model performs well in estimating the increase in population mean mutation rate (Fig 5C). Therefore, mutation rate estimates from previous

studies that neglect heterogeneity in stress-induced mutagenesis, such as [8, 11, 14], can simply be interpreted as population-wide averages. However, these studies may underestimate the true extent of mutagenesis associated with the expression of the stress response if it is only induced by a subpopulation of cells. Estimating not only the increase in population mean but also heterogeneity in mutation rate, as is possible with our method, could be important for parameterising evolutionary models, such as predictions of antibiotic resistance evolution. Theoretical modelling suggests that single-locus adaptation can be accurately captured by the population mean mutation rate, but within-population variation (even for a fixed population mean) can speed up multi-locus adaptation [45]. However, this previous model did not incorporate any coupling of changes in mutation rate to changes in cell division or death rates, as would be expected in the case of stress responses. Therefore, an important direction for future work is to assess when the pleiotropic effects of realistic stress responses truly accelerate evolution.

Our approach to quantifying stress-induced mutagenesis assumes that the expression of the stress response is bimodal and can reasonably be modelled as either switched *off* or *on*. To a reasonable approximation, this expression pattern has been observed for the SOS response, particularly in slow-growth conditions [27]. In other conditions or for other stress responses, it might be more appropriate to model the expression as a unimodal distribution. We expect, however, that this increase in model complexity would make parameter inference more challenging. Similarly, for simplicity, we neglect stochastic induction of the stress response under permissive conditions. Low levels of stress-response expression have been reported, for example, due to spontaneous DNA breakage [46, 47]. We expect our method to be robust to low levels of stress response induction under permissive conditions since, in this case, the subpopulation with elevated stress response level will be negligibly small. This also implies, though, that with our method we cannot effectively quantify heterogeneity in mutation rates in unstressed conditions.

To be able to derive an analytical expression for the mutant count distribution, we made a series of approximations, the most important one being that cells with stress response switched *on* have a net growth rate ($\gamma_{on} - \delta_{on}$) much lower than that of *off* cells ($\gamma_{off}^{s} - \delta_{off} - \alpha$). For the SOS response, this approximation is valid, as induction of the response inhibits cell division. However, it might be violated for other stress responses, particularly if they protect cells from death, resulting in $\delta_{on} < \delta_{off}$. In this case, our approximation is no longer valid, and therefore, parameter estimation using our method is expected to be less accurate. Nonetheless, the estimated mutation-rate increase is robust to relative fitness of response-*on* cells $r_{on} = \frac{\gamma_{on} - \delta_{on}}{\gamma_{off}^{s} - \delta_{off}^{s}}$ up to $\approx$75% and is only marginally improved by inferring $r_{on}$ rather than setting it to zero (Fig 4A).

We also assume response-*on* cells do not switch the response *off* so long as the stress remains present during a fluctuation assay's comparably short growth phase. In particular, this assumption implies that the model cannot capture stress responses that are transiently expressed and associated with pulse-like mutagenesis even under continued exposure to the stressor, such as the oxidative stress response [48]. Our model could be adapted for stress responses where induction of the response is associated with a *decrease* in mutation rate along with increased cell viability, such as the adaptive (Ada) response to DNA alkylation damage, which also exhibits within-population heterogeneity in timing of induction [22, 23]. However, this situation would require a different parameterisation of the model, in which our current analytical approximations break down and the potential for parameter inference would need to be re-tested. Overall, developing models tailored to other stress responses offers an interesting direction for future work.

In summary, we have presented and validated a new method for inferring stress-induced increases in mutation rate from fluctuation assays. Importantly, however, both a heterogeneous stress response and a homogeneous response with mutant fitness costs can generate similar patterns in fluctuation assay data (Fig B in S1 File), which calls for further experiments to distinguish these models. While the homogeneous-response model can estimate the increase in population mean mutation rate, our new method of inferring heterogeneous mutation rates would be crucial for accurately characterising the mutagenic effects of stress responses and parameterising models of multi-locus adaptation. In future work, we aim to incorporate our new method into user-friendly tools for application to experimental data, similar to existing R packages [37, 49] and web tools [16, 32, 50, 51] for fluctuation analysis.

## Supporting information

**S1 File. Supplementary information.** Mathematical derivations, example mutant count distributions, sensitivity analysis, 95% confidence intervals, parameter estimation and model selection for additional parameter ranges, and comparison of model selection procedures. (PDF)

## Acknowledgments

The authors are grateful for helpful feedback on the mathematical results received from Tibor Antal and for the discussion and inspiration provided by the Alexander and El Karoui labs.

## Author Contributions

**Conceptualization:** Lucy Lansch-Justen, Meriem El Karoui, Helen K. Alexander.

**Formal analysis:** Lucy Lansch-Justen.

**Funding acquisition:** Meriem El Karoui, Helen K. Alexander.

**Investigation:** Lucy Lansch-Justen.

**Methodology:** Lucy Lansch-Justen, Meriem El Karoui, Helen K. Alexander.

**Software:** Lucy Lansch-Justen.

**Supervision:** Meriem El Karoui, Helen K. Alexander.

**Validation:** Lucy Lansch-Justen, Helen K. Alexander.

**Visualization:** Lucy Lansch-Justen.

**Writing – original draft:** Lucy Lansch-Justen.

**Writing – review & editing:** Lucy Lansch-Justen, Meriem El Karoui, Helen K. Alexander.

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
