## [Decision Letter · Decision Letter 0]

15 Jan 2024

Dear Ms Lansch-Justen,

Thank you very much for submitting your manuscript "Estimating mutation rates under heterogeneous stress responses" for consideration at PLOS Computational Biology.

As with all papers reviewed by the journal, your manuscript was reviewed by members of the editorial board and by several independent reviewers. In light of the reviews (below this email), we would like to invite the resubmission of a significantly-revised version that takes into account the reviewers' comments.

Both reviewers appreciate the novelty and importance of addressing stress-induced rate heterogeneity in estimating mutation rates. We agree with this evaluation. One of the reviewers raises four major criticisms: 1) that alternative model selection methods could be used, 2) that your method could be applied to empirical data, 3) that confidence intervals should be supplied for the estimated parameters and 4) that the manuscript could be better organised and presented.

The last two points should be straightforward to address, and both reviewers have provided many suggestions for improving the manuscript. To address the first point you can consider whether cross-validation or AIC is the preferred approach; please justify your ultimate choice at least in your response -- and in the revised manuscript if you judge it to be appropriate to do so. On the second point, relating to using empirical data, we agree that such an application would enhance the paper. If it is feasible to do this, it would be a nice addition; otherwise, again, please explain and justify why it cannot be done in this paper.

We cannot make any decision about publication until we have seen the revised manuscript and your response to the reviewers' comments. Your revised manuscript is also likely to be sent to reviewers for further evaluation.

Sincerely,

Mark M. Tanaka

Academic Editor

PLOS Computational Biology

Denise Kühnert

Section Editor

PLOS Computational Biology

Both reviewers appreciate the novelty and importance of addressing stress-induced rate heterogeneity in estimating mutation rates. We agree with this evaluation. One of the reviewers raises four major criticisms: 1) that alternative model selection methods could be used, 2) that your method could be applied to empirical data, 3) that confidence intervals should be supplied for the estimated parameters and 4) that the manuscript could be better organised and presented.

The last two points should be straightforward to address, and both reviewers have provided many suggestions for improving the manuscript. To address the first point you can consider whether cross-validation or AIC is the preferred approach; please justify your ultimate choice at least in your response -- and in the revised manuscript if you judge it to be appropriate to do so. On the second point, relating to using empirical data, we agree that such an application would enhance the paper. If it is feasible to do this, it would be a nice addition; otherwise, again, please explain and justify why it cannot be done in this paper.

Reviewer's Responses to Questions

**Comments to the Authors:**

Reviewer #1: The manuscript by Lansch-Justen et al. describes theory and modelling approaches to estimate mutation rates in bacteria. The work is novel and important because it addresses a crucial gap in the theoretical tools available to model and quantify mutagenesis. Specifically, existing methods to estimate mutation rates assume that all cells in a population have the same mutation rate. However, recent experimental studies have shown that this is not the case, especially under stress conditions when mutation rates increase heterogeneously in a bacterial population. Fluctuation tests are the most common technique to estimate mutation rates in the lab. The method is seemingly straightforward but quantification of the data is mathematically surprisingly complex. For example, it is unclear whether the mutant count distribution obtained from a fluctuation test can tell whether mutation rates vary between cells, and how reliably various parameters can be estimated from the measured data. This work describes a simple model where a population of bacteria splits into two subpopulations with different mutation rates, fitness, and viability in response to stress. Equations are derived to estimate these parameters from data of simulated fluctuation tests. The results show under what conditions model inference and parameter estimation are reliable and when they are not. This provides a useful theoretical footing for the design of future experimental studies. I suggest addressing the following minor points before publication:

1. Abstract: “Moreover, we find that in many cases, our model of heterogeneous stress responses and the standard model with mutant fitness cost reproduce fluctuation assay data equally well, suggesting that separate experiments would be required to identify the true underlying process.” I suspect that many readers would struggle to understand this sentence since it’s not necessarily well known what “the standard model with mutant fitness cost” is. I found the Author Summary more informative than the Abstract.

2. Line 53: specify error-prone “DNA” polymerases

3. Line 90 following: “Single-cell studies using fluorescent reporters for the SOS response in E. coli have revealed that its expression is highly heterogeneous and that this heterogeneity can be approximated as a bimodal response, with some cells expressing low levels of SOS under DNA damage whilst others show a very high level of expression”. It is debatable whether SOS induction is bimodal; studies that reported distributions of SOS expression levels showed a unimodal population with a tail rather than bimodal populations. The discrete on/off model is a reasonable simplification for the purpose of this study (to keep it mathematically tractable), but it should be acknowledged that it doesn’t match the experimental data precisely.

4. Effect of stress response on cell division rate: The SOS response is known to inhibit cell division while cell growth continues, leading to cell filamentation. This should be mentioned in the manuscript and the effects on mutation rate estimate discussed (especially since the authors have studied this effect before https://pubmed.ncbi.nlm.nih.gov/29470493/)

5. The model considers the case where cells switch from the response-off to response-on state during stress. It would be useful to discuss to what extent the results can be extrapolated to situations where cells switch back to the response-off state (relevant for e.g. recovery after stress removal, or stochastic switching between on and off states during unstressed conditions or low stress levels).

6. “Section 2.2.1 Inferred and measured parameters” was a bit confusing to read because it repeats almost the same text for the heterogeneous and homogeneous models. This could be improved by describing the common parameters first, and then noting the differences between the two models?

7. Line 405: “For each parameter combination, we simulate R = 100 pairs of fluctuation assays under permissive and stressful conditions, with c = 50 parallel cultures per assay.” How sensitive are the parameter estimation and model inference results described here to the number of assays and cultures per assay? Can this be added as a supplementary figure? It would provide useful guidance for the design of experiments. Also, how do these values compare to what is typically done in the lab? It would be good to cite some papers here to justify the chosen values.

8. Line 412: “beginning with the simplest model of heterogeneous stress responses without cell death, with mutant fitness equal to non-mutant fitness, and with zero division rate of response-on cells.” I was wondering why the authors didn’t start with the simpler model of homogenous stress response for simplicity, and subsequently move on to the more complex heterogeneous stress response case?

9. Line 413: The case of zero division rate of response-on cells seems rather extreme. If the response-on cells do not divide at all, then shouldn’t this cancel much of the increase in their mutation rate? I probably missed something here, but it could be helpful to explain this point more clearly.

10. It would be useful to include a figure before figure 2 with representative model outputs that show the number of cells in the response-on and response-off populations vs time along with the number of mutants vs time. This could be shown for a few different parameter sets and model types (homogenous, heterogeneous, with/without fitness differences, with/without death, etc). This should give the reader a better intuition for the more abstract results about the inference method performance.

11. Another relevant paper to cite: https://pubmed.ncbi.nlm.nih.gov/30409883/

12. Note that the situation for alkylation stress response in reference 20 actually matches the described model quite well, but in the opposite way. Here, the response-off population has a higher mutation rate and lower survival than the response-on population. To what extent can the theoretical findings be applied to this opposite case?

Reviewer #2: I have read the manuscript titled “Estimating mutation rates under heterogeneous stress responses”.

The authors developed a population dynamic model and maximum likelihood approach to estimate mutation rate heterogeneity from fluctuation assays. This is an important first step towards this goal: evidence for mutation rate heterogeneity (i.e., phenotypic plasticity), especially under stress (González et al 2008) but even without it (Uphoff et al 2016), has been accumulating. I therefore salute the authors for this first step.

• I have some issues that I think the authors can help resolve. Some are related to the presentation– I think the methods section can be organized better and I think some of the methods are not explicitly explained, although I might have missed the explanation (which leads back to better organization…).

• I think the model selection section parts might be enhanced is other methods (CV or LRT) are used.

• I wonder if the authors could have applied their method on some previously published dataset (many fluctuation assays have been published!) and thus demonstrate its usefulness on real data. I appreciate that this might not be feasible because of the requirements of this new method, but in that case I would like to read a clear explanation of what kind of experimental setup is required to use the new method.

• Finally, I think the authors should consider if their method can produce some kind of confidence intervals around the maximum likelihood point estimates; biologists usually want to see something like that when they estimate continuous quantities.

More details on these comments, and more comments, including very minor ones, are listed below.

Major comments:

1. Have you considered applying your method on available data? Maybe even the data from the original Luria-Delbrück paper? See for example Holmes et al, Physical Biology 2017 https://doi.org/10.1088/1478-3975/aa8230. It could be interesting to see (1) if your method performs well on real data; and (2) if your method is able to detect heterogeneity in data for which no heterogeneity has been previously reported.

2. I think that two subpopulations with different mutation rates can also occur without stress response, for examples see theoretical study by Lobinska et al Genetics 2023 and refs within. Does your method accommodate such cases?

3. I think it would have been better to first describe the full model and then describe in a separate section the simple model used for inference – and make it explicit using equations which assumptions you make (e.g. do you set rho=1? In one point you make it explicit in line 222 but I would appreciate making things more explicit in other places, too) and where stochastic processes are involved. The way things are, it is confusing to me and I doubt I could reproduce either model just from the text.

4. It is especially unclear to me how the stochastic simulation works. You mention in some places a branching process but I don’t think I saw a clear and formal description on how it works – the equations seem to focus on a deterministic model and the approximations required for the max likelihood estimation. Do you use Gillespie algorithm? Moran process? SDE? Something else?

5. I think the methods section should start with a clear definition of how the fluctuation assay works, as it is not explicitly defined. There’s a growth phase (which is standard fluctuation tests does not include selection but to infer SIM it does) followed by a selection phase in which mutants that occurred in the growth phase are selected using some stressor (the original Luria-Delbruck used phages right? And now days people usually used some rifampicin and other drugs?). I’m not sure I managed to understand all the details. Maybe even add an illustration of the experiment…?

6. Model selection - it seems strange to me that you use AIC when you can use Cross-Validation: you use simulations so you have as much data as you want; and even when using real experimental data the experimenter is likely to have dozens of replicates. See Arlot & Celisse 2010 10.1214/09-SS054. I would at least like to see if another model selection approach performs better than AIC or if it is just not possible to select between the models (which may be the case, unfortunately).

7. Model selection 2- is the homo-response model not nested within the hetero-response? That is, if you set parameters of the hetero-response model to some fixed values, can you get to the homo-response model? Maybe f_on=0…? If so, you could use the likelihood-ratio test (Wilks test), which in many cases is stronger than information criteria like AIC.

8. Did you check what happens if you simulate from the homo-response model and infer both homo- and hetero-response models? In that case, what is the rate at which you will select the wrong (hetero) model?

9. When the method fails to choose the correct (hetero) model, is it because you infer f_on? if you use the correct f_on rather than inferring it, thus having less parameters (k), will you make the correct model choice?

10. You perform the estimation on two datas

---

## [Decision Letter · Decision Letter 1]

8 May 2024

Dear Ms Lansch-Justen,

We are pleased to inform you that your manuscript 'Estimating mutation rates under heterogeneous stress responses' has been provisionally accepted for publication in PLOS Computational Biology.

Best regards,

Mark M. Tanaka

Academic Editor

PLOS Computational Biology

Denise Kühnert

Section Editor

PLOS Computational Biology

Reviewer's Responses to Questions

**Comments to the Authors:**

Reviewer #1: All my comments have been addressed

Reviewer #2: The authors have responded to my comments in a clear and efficient manner. I think this paper is a fine contribution to the literature and I am looking forward to seeing it published. I have no further comments

**Have the authors made all data and (if applicable) computational code underlying the findings in their manuscript fully available?**

Reviewer #1: None

Reviewer #2: Yes

PLOS authors have the option to publish the peer review history of their article (what does this mean?). If published, this will include your full peer review and any attached files.

Reviewer #1: No

Reviewer #2: **Yes: **Yoav Ram

---

## [Editor Report · Acceptance letter]

23 May 2024

PCOMPBIOL-D-23-01912R1 

Estimating mutation rates under heterogeneous stress responses

Dear Dr Lansch-Justen,

I am pleased to inform you that your manuscript has been formally accepted for publication in PLOS Computational Biology. Your manuscript is now with our production department and you will be notified of the publication date in due course.

With kind regards,

Anita Estes
